# Clinically Important Phleboviruses and Their Detection in Human Samples

**DOI:** 10.3390/v13081500

**Published:** 2021-07-30

**Authors:** Amy J. Lambert, Holly R. Hughes

**Affiliations:** 1Arbovirus Reference and Reagent Laboratory, WHO Collaborating Center, Division of Vector Borne Diseases, Centers for Disease Control and Prevention, 3156 Rampart Road-Foothills Research Campus, Fort Collins, CO 80521, USA; ltr8@cdc.gov; 2Arbovirus Diagnostic and Reference Team, Division of Vector Borne Diseases, Centers for Disease Control and Prevention, 3156 Rampart Road-Foothills Research Campus, Fort Collins, CO 80521, USA

**Keywords:** phleboviruses, viral detection, human samples

## Abstract

The detection of phleboviruses (family: *Phenuiviridae*) in human samples is challenged by the overall diversity and genetic complexity of clinically relevant strains, their predominantly nondescript clinical associations, and a related lack of awareness among some clinicians and laboratorians. Here, we seek to inform the detection of human phlebovirus infections by providing a brief introduction to clinically relevant phleboviruses, as well as key targets and approaches for their detection. Given the diversity of pathogens within the genus, this report focuses on diagnostic attributes that are generally shared among these agents and should be used as a complement to, rather than a replacement of, more detailed discussions on the detection of phleboviruses at the individual virus level.

## 1. An Introduction to Clinically Important Phleboviruses 

As of this writing, 11 phlebovirus species isolated from geographic locations spanning both hemispheres are associated with human disease (Table 1). This number is subject and likely to change due to an evolving taxonomy [1], as well as the remarkable rate of recent phlebovirus discoveries, such as Ntepe and Drin viruses [2,3] and the detection of novel reassortant viruses, such as Ponticelli I, II, and III in the arthropod host [4]. Briefly, and as discussed elsewhere in this issue [1], reassortant phleboviruses result from an exchange of genomic segments between related parental phlebovirus strains. This phenomenon is facilitated by the segmented nature of the tripartite phlebovirus genome and could predicate novel disease emergence should that exchange of segments confer some fitness advantage or altered pathogenicity in the human host. Therefore, the ability to detect and identify reassortant viruses is of special clinical, epidemiological, and public health interest.

The majority of phlebovirus strains are maintained and transmitted by phlebotomine sandflies. While most infections are thought to be asymptomatic, the typical “sandfly fever” symptoms include the sudden onset of fever, malaise, anorexia, photophobia, abdominal symptoms, and rash [5,6,7]. These symptoms are generally associated with Old World (Sandfly fever Naples and Sicilian) and New World (Alenquer, Candiru, Chagres, Cocle, Echarate, Maldonado, and Punta Toro) phleboviruses (Table 1). Similarly, infections with the mosquito-borne Rift Valley fever virus are most often associated with a self-limiting febrile illness [5,6]. Unfortunately, a small subset of Rift Valley fever virus human cases can progress into hemorrhagic fever, hepatitis, encephalitis, and/or retinal vasculitis [5,6,7], representing the most severe human clinical manifestations associated with a phlebovirus infection. Of special interest, Rift Valley fever virus is also known to cause high rates of mortality and abortion among infected livestock, with epizootics occurring along with the development of illness in the people who tend these animals [5,6,8]. Lastly, Toscana virus is the only sandfly-borne phlebovirus that is frequently associated with aseptic meningitis [7] in addition to a more common febrile syndrome. This unique presentation facilitates the diagnosis of Toscana virus infections in the clinical setting, particularly in Italy during the summer months where physicians are aware of its likely circulation and distinguishing (among sandfly-borne phleboviruses) disease association.

## 2. Key Targets for the Detection of Phlebovirus Infections

The interplay of viremia and the host immune response determines the window of opportunity and targets (virus or antibodies) for diagnosis of all viral infections. For phlebovirus infections, both virus (whole virus, antigen, nucleic acid) and immune (IgM, IgG, neutralizing antibodies) components are useful targets for diagnosis [7,9]. However, an exact determination of what target(s) is/are best at what time after the onset of illness has not been systematically derived for most implicated phlebovirus strains given their orphan, neglected status. In general, whole virus, nucleic acid, and antigens are most likely to be detected within the first few days of febrile illness when viremia is high [7,9], with waning and more sporadic utility thereafter. Inference of phlebovirus infections through the detection of antibodies can occur for a broader window of time. IgM is generally detectable very early within the first week after the onset of illness and continues to be detectable for weeks or months thereafter [9], making IgM an excellent target for inference of acute infection [7]. IgG and neutralizing antibodies rise within the first several weeks [9,10] and are detectable for years after infection, making these antibodies outstanding markers of seroprevalence [11,12,13,14]. In general, a four-fold or greater rise in antibody titer between paired sera is diagnostic of acute infection [9]. Human serum and CSF are the most common sample types subjected to analyses; however, postmortem tissues, whole blood, and urine may also be of use for direct detection methods, in particular [9,15,16]. 

## 3. Methods for the Detection of Phleboviruses and Their Infections

### 3.1. Direct Detection 

Classical methods for the discovery and detection of phleboviruses include isolation by inoculation of either suckling mice or susceptible cells (e.g., Vero cells) with sera, CSF samples, or supernatants of homogenates derived from tissues of infected individuals or arthropods [7,9]. Following isolation, identification and characterization of newly derived isolates were formerly provided by predominantly antibody-based methods, including complement fixation (CF), hemagglutination inhibition (HI), immunofluorescence assays (IFAs), and plaque reduction neutralization tests (PRNTs) [7,17]. In recent years, isolates have become increasingly characterized by nucleic-acid-based methods [18,19,20], including whole-genome sequencing, rather than serology. This transition has facilitated the more rapid identification of reassortant viruses [4,21] and allows for taxonomic classification based upon nucleic-acid-based criteria for demarcation [1]. In fact, with the advent of RT-PCR, isolation-based methods have become more infrequently used altogether in the interest of the relatively fast, specific answer that these methods, including nested, real-time, and consensus formats, can provide when directly applied to clinical samples [22,23,24,25]. Consensus RT-PCR assays detecting the small segment [24] or utilizing a nested approach to detect both small and large segments [22], have been particularly useful for the detection of a broad diversity of species in the context of clinical and outbreak investigations, virus discovery, and surveillance studies [26,27,28]. These assays are designed to detect a group of viruses of interest, followed by nucleotide sequencing for result confirmation and virus identification. When targeting multiple segments or when used in combination with, rather than in replacement of, virus isolation, serology, and full-genomic sequencing, these methods also rapidly facilitate the detection of reassortant strains [29]. 

### 3.2. Detection of Antibodies

Serological inference of phlebovirus infections was historically provided primarily through the detection of antibodies by HI and PRNT evaluations of serum samples [7,11]. With the development of monoclonal antibodies (MAbs), the enzyme-linked immunosorbent assay (ELISA), and recombinant antigens, serology for all viral infections has become more broadly used for frontline acute diagnosis. ELISA and IFA assays for the detection of IgM and IgG in both kit and in-house forms are frequently utilized during phlebovirus serosurveys [13,15,30]. In addition, there is also growing evidence that serological approaches for the inference of phlebovirus infections are surprisingly specific and offer more sensitivity for detection of acute infections than previously known [30]. This high level of specificity was demonstrated when the newly identified Ponticelli I, II, and III viruses were not neutralized by patient serum specimens with confirmed Sandfly fever virus and Toscana virus antibodies [30]. Specificity is enhanced if frontline IgM and IgG screening are complemented by PRNT confirmatory analyses, demonstrating a seroprevalence of Sandfly fever infections as high as 42% in one study [30], additionally speaking to the likely underestimation of phleboviruses in the clinical setting. Undoubtedly, the availability of a full repertoire of direct detection and antibody-based methods is the best way to ensure the probability of detecting a phlebovirus infection in the human host. 

### 3.3. Historical Impact, Continued Emergence and Recommendations for Future Detection and Discovery

Outbreaks of human disease have been both contemporaneously and retrospectively associated with phlebovirus infections dating back to Napoleonic times [31,32,33]. Of note, sandfly fever has been alternatively and historically referred to as “pappataci”, translated roughly from Italian as “to eat silently”, fever referring to the cryptic feeding habits of phlebotomine flies, or “three-day fever”, describing the self-limiting febrile illness associated with most pathogenic phleboviruses. Phleboviruses were also responsible for a significant troop morbidity, as primarily documented in the Mediterranean theater during the Second World War [31]. In this context, phleboviruses were also likely responsible for some cases of “trench fever”, more commonly associated with louse-borne rickettsial disease, on the limited Mediterranean front in the First World War. This deep history, along with increasing globalization and continued disease emergence [27,34,35], tells us that phleboviruses will be clinically important on a broader geographic scale well into the future. Accordingly, and as informed by our own collaborative experiences [26,27,36,37], we propose that broadly reactive consensus and nested RT-PCR approaches to phlebovirus detection are ideal methods to potentiate the discovery of viruses of novel circumstance and description, including reassortant strains. While limited in their utility to the very acute phase of infection, their design allows for the sensitive detection of a broad diversity of known and potentially unknown, but genetically related, agents. When complemented by a broad repertoire of approaches, including virus isolation, serological and full-genomic sequencing methods, these consensus assays have provided us and others with a great “first shot” of detecting emerging pathogens by molecular means without species level a priori knowledge of the infectious agent [26,27,38,39].

## Figures and Tables

**Table 1 viruses-13-01500-t001:** Known human pathogens of the genus *Phlebovirus* and typical associations *.

Type Species Common Name	Virus Strains ^	Disease(s)	Vector/Mode of Transmission	Isolated From
Alenquer		Self-limiting fever °	Unknown	Brazil
Candiru	Candiru virus	Self-limiting fever	Unknown	Brazil
	Morumbi virus	Self-limiting fever	Unknown	Brazil
	Serra Norte virus	Self-limiting fever	Unknown	Brazil
Chagres		Self-limiting fever	Sandfly	Panama
Cocle		Self-limiting fever	Sandfly	Panama
Echarate		Self-limiting fever	Unknown	Peru
Maldonado		Self-limiting fever	Unknown	Peru
Punta Toro		Self-limiting fever	Sandfly	Panama
Rift Valley fever		Fever, hemorrhagic fever, encephalitis, hepatitis *	Mosquito/aerosol	Africa
Sandfly fever Naples	Sandfly fever Naples virus	Self-limiting fever	Sandfly	Europe, Africa, Asia
	Granada virus	Self-limiting fever	Sandfly	Europe
Toscana		Fever, aseptic meningitis	Sandfly	Mediterranean Europe and Africa
Sandfly fever Sicilian	Sandfly fever Sicilian virus	Self-limiting fever	Sandfly	Europe, Africa, Asia
	Sandfly fever Cyprus virus	Self-limiting fever	Sandfly	Mediterranean Europe
	Sandfly fever Turkey virus	Self-limiting fever	Sandfly	Turkey

* Rift Valley fever virus is also a known veterinary pathogen that is associated with high rates of mortality and abortion in livestock. ^ Only strains that have been directly associated with human disease are listed. ° Commonly, but not exclusively, of 3 days duration and marked by fatigue, muscle and joint pain, headache, and nausea.

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
