# Peer review of "Clinically Important Phleboviruses and Their Detection in Human Samples"

_viruses, 2021, doi:10.3390/v13081500_

Round 1

Reviewer 1 Report

This is a very short review of clinically relevant phleboviruses and some general considerations on the laboratory diagnosis of these infections. The manuscript provides a brief description of the clinical syndromes associated with phlebovirus infection in humans and an overview of diagnostic methods and the kinetics of diagnostic markers. The text is well written and informative, based on the literature and the author’s experience in the field.

The lack of clinical suspicion and testing may contribute to a lower detection of human phlebovirus infections and therefore underestimate the real magnitude of phlebovirus infections. Some additional information in the manuscript could help the reader to understand that phlebovirus infections may be widespread and may increase the interest in testing compatible clinical syndromes:

  1. Specific examples of newly described viruses, even if they have not been associated yet with human disease but have been detected in vectors.
  2. Information on seroprevalence studies. Some studies show high prevalence of antibodies against phleboviruses This suggests that infections are underdiagnosed / some are asymptomatic or mid but in any case that the distribution and circulation of these viruses is likely higher than initially expected.
  3. More information on consensus molecular methods that are recommended for phlebovirus investigation i.e. genomic regions targeted by these methods

Other minor comments:

  1. Table 1. Toscana virus infections not only cause CNS infections. It should be indicated that they also cause a febrile syndrome. Febrile illness is even more frequent than neuroinvase infections.
  2. It should be mentioned that phlebovirus infections can be also asymptomatic.
  3. It is written: “In addition, there is also growing evidence that serological approaches for the inference of phlebovirus infections are surprisingly specific and offer more sensitivity for detection of acute infections than previously known.” This may be a bit ambiguous. In the reference 31 they show that more patients were diagnosed by serological methods than by molecular methods, which is probably the case for most arboviral infections. The sensitivity and specificity of the serological diagnosis will largely depend on the particular test used (IFA /ELISA/ in house/commercial) and the use of neutralization tests, which are not available in most clinical microbiology laboratories. Serological cross reactivity with antibodies against other phleboviruses can occur with screening methods and even with neutralization assays.

Author Response

Please see our itemized responses to the Reviewers in italics, below.

Reviewer 1

This is a very short review of clinically relevant phleboviruses and some general considerations on the laboratory diagnosis of these infections. The manuscript provides a brief description of the clinical syndromes associated with phlebovirus infection in humans and an overview of diagnostic methods and the kinetics of diagnostic markers. The text is well written and informative, based on the literature and the author’s experience in the field.

The lack of clinical suspicion and testing may contribute to a lower detection of human phlebovirus infections and therefore underestimate the real magnitude of phlebovirus infections. Some additional information in the manuscript could help the reader to understand that phlebovirus infections may be widespread and may increase the interest in testing compatible clinical syndromes:

  1. Specific examples of newly described viruses, even if they have not been associated yet with human disease but have been detected in vectors.
    • Specific names of referenced new viruses have now been included in the introduction text, lines 26-27.
  2. Information on seroprevalence studies. Some studies show high prevalence of antibodies against phleboviruses This suggests that infections are underdiagnosed / some are asymptomatic or mild but in any case that the distribution and circulation of these viruses is likely higher than initially expected.
  • Lines 118-121 now include additional information on the high prevalence of sandfly fever virus infections to support this point.
  1. More information on consensus molecular methods that are recommended for phlebovirus investigation i.e. genomic regions targeted by these methods
  • Lines 95-98 now include additional information on consensus approaches

Other minor comments:

  • Table 1. Toscana virus infections not only cause CNS infections. It should be indicated that they also cause a febrile syndrome. Febrile illness is even more frequent than neuroinvase infections.
    • Line 56 and Table 1 now include this
  • It should be mentioned that phlebovirus infections can be also asymptomatic.
  • This is now included in line 43
  • It is written: “In addition, there is also growing evidence that serological approaches for the inference of phlebovirus infections are surprisingly specific and offer more sensitivity for detection of acute infections than previously known.” This may be a bit ambiguous. In the reference 31 they show that more patients were diagnosed by serological methods than by molecular methods, which is probably the case for most arboviral infections. The sensitivity and specificity of the serological diagnosis will largely depend on the particular test used (IFA /ELISA/ in house/commercial) and the use of neutralization tests, which are not available in most clinical microbiology laboratories. Serological cross reactivity with antibodies against other phleboviruses can occur with screening methods and even with neutralization assays.
  • Lines 115-117 now describe in more depth a lack of cross-neutralization between viruses of the Sandfly Naples species complex

Reviewer 2

Review: Lambert, Hughes, phleboviruses

Comments:

  1. Abstract, line 11: What does “their overall diversity” refer to, the viruses or the human samples?
  2. Line 15: It would be useful to spell out the taxonomy of these viruses before using “the genus”.
  3. Line 16: “agents and which should be”
  4. Line 18: Replace “species” with “virus” because no method can be used to identify a species, which are non-concrete entities.
  5. Why in Table 1, do the words “sandfly fever” appear so often? It is difficult for me to understand why they are in the table.
  6. “Sicilian” is misspelled
  7. Last line of Table 1: “Turkey” is misspelled once.
  8. Footnote to Table 1: “Rift Valley fever” (lower case “fever”)
  9. Line 69: “add weeks or months” to “months”. Weeks could be 200 weeks and months could be 200 months.  Usually, the IgM response can be detected for a limited time after it is first detected.  That is why the IgM assay is so useful in detecting recent infections.
  10. Line 72: “four-fold or greater rise”
  11. Line 72: Not “in” paired sera, “between” paired sera.
  12. Line 74: Should there not be a semicolon before “however”?
  13. Line 83: One “m” in “hemagglutination”
  14. Line 96: Replace “speciation” with “identification”
  15. Line 105: “well reported” seems an awkward phrase. What does it mean?
  16. Lines 117-120: “silent porridge” is the worst translation of pappataci I have ever seen. The name pappataci fever comes from the Italian word for sandfly; it is the union of the word "pappa" (food) and taci (silent), indicating that the little buggers sneak up on you and feed on you when you least expect it.  Sandflies do not make the buzzing sounds that mosquitoes do.
  17. References:
  18. The left margins are not all aligned.
  19. Reference 11: Period after “Bull” (and that’s no bull)
  20. References 4 and 21 are the same. Delete one and renumber references.
  21. Reference 26: Why the underlining?
  22. Reference 32: Remove parentheses from date
  23. Check to be certain that words in the original references are not italicized (such as the species names of mosquitoes, etc.)

Otherwise, besides these trivial comments, this is a nice little manuscript; useful also.

Response to Reviewer 2, comments 1-23:  All of the suggested edits and clarifications have been incorporated into the revised text and table.

Reviewer 2 Report

//

Review: Lambert, Hughes, phleboviruses

Comments:

  1. Abstract, line 11: What does “their overall diversity” refer to, the viruses or the human samples?
  2. Line 15: It would be useful to spell out the taxonomy of these viruses before using “the genus”.
  3. Line 16: “agents and which should be”
  4. Line 18: Replace “species” with “virus” because no method can be used to identify a species, which are non-concrete entities.
  5. Why in Table 1, do the words “sandfly fever” appear so often? It is difficult for me to understand why they are in the table.
  6. “Sicilian” is misspelled
  7. Last line of Table 1: “Turkey” is misspelled once.
  8. Footnote to Table 1: “Rift Valley fever” (lower case “fever”)
  9. Line 69: “add weeks or months” to “months”. Weeks could be 200 weeks and months could be 200 months.  Usually, the IgM response can be detected for a limited time after it is first detected.  That is why the IgM assay is so useful in detecting recent infections.
  10. Line 72: “four-fold or greater rise”
  11. Line 72: Not “in” paired sera, “between” paired sera.
  12. Line 74: Should there not be a semicolon before “however”?
  13. Line 83: One “m” in “hemagglutination”
  14. Line 96: Replace “speciation” with “identification”
  15. Line 105: “well reported” seems an awkward phrase. What does it mean?
  16. Lines 117-120: “silent porridge” is the worst translation of pappataci I have ever seen. The name pappataci fever comes from the Italian word for sandfly; it is the union of the word "pappa" (food) and taci (silent), indicating that the little buggers sneak up on you and feed on you when you least expect it.  Sandflies do not make the buzzing sounds that mosquitoes do.
  17. References:
  18. The left margins are not all aligned.
  19. Reference 11: Period after “Bull” (and that’s no bull)
  20. References 4 and 21 are the same. Delete one and renumber references.
  21. Reference 26: Why the underlining?
  22. Reference 32: Remove parentheses from date
  23. Check to be certain that words in the original references are not italicized (such as the species names of mosquitoes, etc.)

Otherwise, besides these trivial comments, this is a nice little manuscript; useful also.

Author Response

(The authors gave the same response as above.)
